# DNA Methylation Patterns Correlate with the Expression of *SCNN1A*, *SCNN1B*, and *SCNN1G* (Epithelial Sodium Channel, ENaC) Genes

**DOI:** 10.3390/ijms22073754

**Published:** 2021-04-04

**Authors:** Silvia Pierandrei, Gessica Truglio, Fabrizio Ceci, Paola Del Porto, Sabina Maria Bruno, Stefano Castellani, Massimo Conese, Fiorentina Ascenzioni, Marco Lucarelli

**Affiliations:** 1Department of Experimental Medicine, Sapienza University of Rome, Viale Regina Elena 324, 00161 Roma, Italy; pierandrei.silvia@gmail.com (S.P.); gessica.truglio@gmail.com (G.T.); fabrizio.ceci@uniroma1.it (F.C.); sabinamaria.bruno@gmail.com (S.M.B.); 2Department of Biology and Biotechnology “Charles Darwin”, Sapienza University of Rome, Via dei Sardi 70, 00185 Roma, Italy; paola.delporto@uniroma1.it; 3Department of Biomedical Sciences and Human Oncology, University of Bari, Piazza Giulio Cesare 11, 70124 Bari, Italy; stefano.castellani@uniba.it; 4Department of Medical and Surgical Sciences, University of Foggia, Via Napoli 121, 71122 Foggia, Italy; massimo.conese@unifg.it; 5Pasteur Institute, Cenci Bolognetti Foundation, Sapienza University of Rome, Viale Regina Elena 291, 00161 Roma, Italy

**Keywords:** epithelial sodium channel, DNA methylation, transcriptional control, cystic fibrosis

## Abstract

The interplay between the cystic fibrosis transmembrane conductance regulator (CFTR) and the epithelial sodium channel (ENaC) in respiratory epithelia has a crucial role in the pathogenesis of cystic fibrosis (CF). The comprehension of the mechanisms of transcriptional regulation of ENaC genes is pivotal to better detail the pathogenic mechanism and the genotype–phenotype relationship in CF, as well as to realize therapeutic approaches based on the transcriptional downregulation of ENaC genes. Since we aimed to study the epigenetic transcriptional control of ENaC genes, an assessment of their expression and DNA methylation patterns in different human cell lines, nasal brushing samples, and leucocytes was performed. The mRNA expression of *CFTR* and ENaC subunits α, β and γ (respectively *SCNN1A*, *SCNN1B,* and *SCNN1G* genes) was studied by real time PCR. DNA methylation of 5′-flanking region of *SCNN1A*, *SCNN1B,* and *SCNN1G* genes was studied by HpaII/PCR. The levels of expression and DNA methylation of ENaC genes in the different cell lines, brushing samples, and leukocytes were very variable. The DNA regions studied of each ENaC gene showed different methylation patterns. A general inverse correlation between expression and DNA methylation was evidenced. Leukocytes showed very low expression of all the 3 ENaC genes corresponding to a DNA methylated pattern. The *SCNN1A* gene resulted to be the most expressed in some cell lines that, accordingly, showed a completely demethylated pattern. Coherently, a heavy and moderate methylated pattern of, respectively, *SCNN1B* and *SCNN1G* genes corresponded to low levels of expression. As exceptions, we found that dexamethasone treatment appeared to stimulate the expression of all the 3 ENaC genes, without an evident modulation of the DNA methylation pattern, and that in nasal brushing a considerable expression of all the 3 ENaC genes were found despite an apparent methylated pattern. At least part of the expression modulation of ENaC genes seems to depend on the DNA methylation patterns of specific DNA regions. This points to epigenetics as a controlling mechanism of ENaC function and as a possible therapeutic approach for CF.

## 1. Introduction

Pathogenic variants of the Cystic Fibrosis Transmembrane conductance Regulator (*CFTR*) gene, by lowering or abolishing the CFTR Cl^-^ channel activity, cause a broad spectrum of clinical manifestations spanning from Cystic Fibrosis (CF) to the so-called CFTR related disorders (CFTR-RD) [1,2]. It has been suggested that dysfunctional CFTR impacts the activity of epithelial sodium channel (ENaC) causing an altered interaction between CFTR and ENaC that, in respiratory epithelia, contributes to both CF and CFTR-RD [3,4,5,6,7,8]. Wild-type CFTR inhibits ENaC function by impeding its proteolytic activation and reducing channel opening [9,10,11]. On the contrary, mutated CFTR is not able to protect ENaC from proteolytic cleavage and activation [10], which, in turn, results in higher Na^+^ absorption by epithelial cells. Through the Na^+^ transport across respiratory epithelium, the ENaC regulates hydration of the airways surface liquid [12,13]. Its expression in lungs has a crucial role for physiologic pulmonary function and regeneration [12]. Therefore, the coordinated transport of Cl^-^ and Na^+^ mediated by CFTR and ENaC contributes to the correct hydration of human airways [14,15,16]. In addition, this functional network originates a complex relationship between genotype and phenotype in CF [17,18,19,20] and impact on its diagnosis, prognosis, and therapy [2,21,22].

ENaC is composed of the α, β and γ subunits, coded by the genes for the sodium channel epithelial 1 subunits α, β and γ, respectively *SCNN1A* [23], *SCNN1B,* and *SCNN1G* [24], whose expression appears to be under the transcriptional control of DNA methylation. The *SCNN1A* gene has a high density of CpG sites [25], whereas *SCNN1B* and *SCNN1G* genes have, respectively, one [26] and two [27,28] CpG islands. In effect, DNA methylation-dependent transcriptional changes of the *SCNN1B* [29,30,31] and *SCNN1G* [28] gene have been described, suggesting that this regulatory mechanism can be targeted to control ENaC expression and activity. Indeed, the targeting of methylation, at both the DNA [28] (with consequent chromatin remodeling) and protein [32] (affecting Na^+^ transport) level, appears to be a promising approach, although barely explored until now. In general, experimental evidence points to the targeting of ENaC as a potential therapy for CF [33,34,35]. Accordingly, it has been shown that the silencing of the α subunit by lentiviral small hairpin RNA caused a significant reduction of the ENaC activity supporting the notion that gene expression inhibition of one out of the three ENaC subunits is sufficient to dampen ENaC activity [36]. Although other anti-ENaC therapeutic tools have been proposed, most of them failed and new approaches are needed [37,38,39,40,41,42,43,44,45].

With the aim of better understanding the mechanisms of transcriptional regulation of ENaC genes, in this work, we studied the expression level and the associated DNA methylation pattern of *SCNN1A*, *SCNN1B,* and *SCNN1G* genes, in different human cell line, nasal brushing samples, and leukocytes. A better comprehension of the transcriptional regulation mechanisms of these genes may better clarify the genotype–phenotype relationship in CF, enhance our diagnostic and prognostic ability, as well as open the way to a realistic therapeutic intervention based on epigenetic silencing of ENaC genes.

## 2. Results

### 2.1. Expression of ENaC Genes

The cellular models selected in this study (described in Materials and Methods) were first assessed for *CFTR* expression by real time PCR. CFTR expression was high in 16HBE, moderate or low in H441, MCF10A and CFBE, and very low in HaCaT (Appendix A). Dexamethasone treatment increased *CFTR* expression in H441, while it caused a reduction in MCF10A (Appendix A). The analysis of ex vivo samples showed a significant expression of *CFTR* only in nasal brushing and, at a lower extent, in monocytes, whereas in the other peripheral blood cells the expression was low or very low (Appendix A).

Next, we assessed the expression of ENaC genes by real time PCR. The results showed high expression of the *SCNN1A* gene in H441 and HaCaT, moderate expression in MCF10A and 16HBE, whereas CFBE showed low expression (Figure 1A). As expected, dexamethasone treatment increased *SCNN1A* gene expression in both H441 and MCF10A (Figure 1B). The highest level of expression of *SCNN1A* gene was evidenced in nasal brushing, whereas it was moderate or low in leukocytes (Figure 1C).

The gene expression of *SCNN1B* resulted generally lower than *SCNN1A* gene (as can be seen from the extension of *y*-axis in Figure 1D,F). Indeed, *SCNN1B* gene was moderately expressed in H441, very low expressed in MCF10A, 16HBE, and CFBE, and undetectable in HaCat (Figure 1D). Similar to the *SCNN1A* gene, dexamethasone treatment increased the expression of the *SCNN1B* gene in both H441 and MCF10A (Figure 1E). As for the *SCNN1A*, the nasal brushing samples showed the highest expression of *SCNN1B* gene, whereas its expression was almost undetectable in leukocytes (Figure 1F).

The gene expression of *SCNN1G* was lower than *SCNN1B* gene (as can be seen from the extension of *y*-axis in Figure 1G,I). In particular, *SCNN1G* was moderately expressed in H441, low expressed in MCF10A and CFBE, and very low expressed in 16HBE and HaCaT (Figure 1G). An increase in expression of *SCNN1G* gene was induced by dexamethasone in both H441 and MCF10A (Figure 1H). Nasal brushing showed a moderate expression of *SCNN1G* gene, whereas a very low expression was found in leukocytes (Figure 1I).

The results obtained by real time PCR revealed that the H441 cell line showed a good expression of all the 3 ENaC genes, in the following quantitative order *SCNN1A* >> *SCNN1B* > *SCNN1G*. HaCaT showed a high expression of *SCNN1A* gene but a very low (if any) expression of *SCNN1B* and *SCNN1G* genes. Similarly, in the other cell lines (MCF10A, 16HBE, and CFBE), the most expressed ENaC gene was the *SCNN1A* with the expression of *SCNN1B* and *SCNN1G* genes being low or very low. In all the cell lines, the *SCNN1A* gene resulted to be the most expressed, although its expression in CFBE was well below to that in the other cell lines. All the 3 ENaC genes were inducible by dexamethasone in both tested cell lines (H441 and MCF10A). All the 3 ENaC genes were shown to be expressed in nasal brushing in the following quantitative order *SCNN1A* >> *SCNN1B* > *SCNN1G*. The expression of the *SCNN1A* gene in nasal brushing was the highest among all the experimental models and conditions herein tested, while the expression of all the 3 ENaC genes in leukocytes were shown to be well below to that in nasal brushing and cell lines and appreciable only for *SCNN1A* gene.

Data from quantitative real time PCR were also displayed by qualitative endpoint PCR and agarose gel electrophoresis, which are reported in supplementary information (*CFTR*, Appendix A; *SCNN1A*, Appendix A; *SCNN1B*, Appendix A; *SCNN1G*, Appendix A).

### 2.2. Methylation of ENaC Genes

The methylation analysis was performed according to the general organization of 5′-flanking regions of ENaC genes reported in Appendix A. The analysis of the three regions containing CCGG sites (a, b and c) and one control region without CCGG sites (Control) of the 5′-flanking region of the *SCNN1A* gene, showed the “a” region demethylated in all the experimental conditions (Figure 2). In the cell lines (Figure 2A), the “b” and “c” regions resulted demethylated in H441, MCF10A and HaCaT cells, in which *SCNN1A* was highly expressed. Moreover, “b” and “c” regions resulted methylated in CFBE cells, where *SCNN1A* was expressed at very low levels. Also, 16HBE cells showed a methylated status in the “c” region according to their moderate *SCNN1A* gene expression levels. Dexamethasone treatment of H441 and MCF10A cells resulted ineffective on methylation status (Figure 2B), although stimulating *SCNN1A* expression. Results from nasal brushing and leukocytes showed a general methylation of both “b” and “c” regions (Figure 2C). The methylation of the “b” and “c” regions appeared well correlated with the low expression of the *SCNN1A* gene in leukocytes, on the contrary to nasal brushing. Overall, with the exception of nasal brushing and dexamethasone treatment, a good inverse correlation between DNA methylation and expression of the *SCNN1A* gene was evidenced.

The analysis of the four regions containing CCGG sites (a, b, c and d) and one control region without CCGG sites (Control) of the 5′-flanking region of the *SCNN1B* gene showed the “b” region demethylated and the “c” region methylated in all the experimental conditions (Figure 3). In the cell lines (Figure 3A,D), the “a” and “d” regions resulted in a general methylated pattern, well correlated with the low expression of *SCNN1B* gene. The methylated pattern of H441 and MCF10A cells did not change after dexamethasone treatment (Figure 3B,E), despite the increase in *SCNN1B* gene expression. A methylated pattern of “a” and “d” regions (in addition to the “c” region), and a concomitant low expression of *SCNN1B* gene, was evidenced also in nasal brushing and leukocytes. Overall, compared to the *SCNN1A* gene, the *SCNN1B* gene showed a heavier methylated pattern which correlated to a lower expression.

The analysis of the two regions containing CCGG sites (a and b) and one control region without CCGG sites (Control) of 5′-flanking of *SCNN1G* gene showed the “b” region demethylated in all the experimental conditions (Figure 4). In cell lines (Figure 4A,D) the “a” region resulted slightly methylated in the H441, 16HBE, CFBE, and HaCaT cells and completely demethylated in MCF10A. This partially methylated pattern resulted to be well correlated to the low expression of *SCNN1G* gene (comparable to that of *SCNN1B* gene), with the exception of the MCF10A cells. Also, for the *SCNN1G* gene, dexamethasone treatment did not modify the basal methylation pattern of H441 and MCF10A cells (Figure 4B,E), although *SCNN1G* gene expression increased after treatment. A similar partially methylated pattern of the “a” region was shown in nasal brushing and leukocytes (Figure 4C,F), and it was associated with a general low gene expression. Compared to the other two ENaC genes, *SCNN1G* gene showed a methylation pattern intermediate between the *SCNN1A* gene (the least methylated in well expressing conditions) and the *SCNN1B* gene (the most methylated in low expressing conditions).

The cumulative results of *SCNN1A, SCNN1B* and *SCNN1G* gene expression and DNA methylation are schematized in Figure 5A,B,C respectively.

## 3. Discussion

The expression levels of the *SCNN1A*, *SCNN1B*, and *SCNN1G* genes, in the different cellular models herein considered, well correlate with the DNA methylation pattern of the regions we have analyzed. This is demonstrated by the correlation of low expression level/high DNA methylation or high expression level/low DNA methylation. Accordingly, leukocytes showed a very low (or absent) expression of all the three genes and a DNA methylation in the regions examined. In cell lines, in particular the *SCNN1A* gene was shown to be the most expressed and the most demethylated, the *SCNN1B* gene was shown to be the least expressed and the most methylated, and the *SCNN1G* gene was shown to have intermediate levels of expression and methylation. Discrepant results were obtained only after dexamethasone treatment and in nasal brushing. Dexamethasone stimulated the expression of all the three ENaC genes without apparent modulation of the DNA methylation. For the *SCNN1A* gene this result may be explained with the completely demethylated pattern already present in basal conditions in the analyzed regions. However, in the *SCNN1B* and *SCNN1G* genes, a methylated pattern potentially editable was present in untreated cells, but no variation could be detected after dexamethasone treatment. It can be concluded that the mechanism of dexamethasone induction of ENaC gene expression does not involve the modulation of DNA methylation of the studied regions. In nasal brushing, a high expression of *SCNN1A* gene and moderate expression of *SCNN1B* and *SCNN1G* gene were found despite the methylated patterns. This is probably due to a mixed composition of nasal tissue where are present both expressing and non-expressing cell types: the expressing cells responsible for the expression detection and the non-expressing cells for the methylated pattern.

The DNA regions studied for methylation appeared to be not equivalent. One region of *SCNN1A* gene resulted always demethylated in all the samples (being the methylation of the other two regions modulated). In *SCNN1B* one region resulted always demethylated and another region always methylated in all the samples (being the methylation of the other two regions partially modulated, although less than in *SCNN1A* gene). In the *SCNN1G* gene one region was always demethylated and the methylation of the other region modulated. It should be taken into account that, with the method here used for methylation analysis, when the amplicon of a specific region is not present it means that at least one HpaII site is unmethylated. On the contrary, when the amplicon of a specific region is present it means that all the HpaII sites are methylated. The great differences in DNA methylation patterns of different regions found with this approach stimulates a finer analysis at single cytosine level, possibly distinguishing between CpG and non-CpG methylation [46] which was shown to be crucial in other contexts [47,48].

Our results highlight that at least part of the transcriptional control of *SCNN1A*, *SCNN1B* and *SCNN1G* genes is exerted by DNA methylation. This points to epigenetics as a mechanism for the regulation of expression and, finally, activity of ENaC in both physiologic and pathologic contexts. In particular, in CF, a better comprehension of the functional modulation of ENaC could help to clarify the genotype–phenotype relationship, as well as to plan innovative therapeutic approaches. The induction of methylated patterns on ENaC genes could downregulate ENaC activity, partially correcting its dysregulated over-activity due to the loss of CFTR-dependent inhibition. A more focused and high-resolution mapping of methylation of CpG and non-CpG moieties within the control regions of ENaC genes appears to be mandatory.

## 4. Materials and Methods

### 4.1. Cells and Culture Conditions

Cell lines: H441 (ATCC^®^ HTB-174TM, Manassass, VA, USA), human lung epithelial cells; MCF10A (ATCC^®^ CRL-10317) human CF mammary epithelial cells; HaCaT (ATCC^®^ CRL-2309), human keratinocytes; 16HBE14o- (16HBE, wt CFTR) and CFBE41o- (CFBE, homozygous for the CFTR Phe508del pathogenic variant), human bronchial epithelial cells kindly provided by Dr Dieter Gruenert [49]. The cell lines were cultured according to supplier’s specifications with minor modifications. Briefly, H441 cells were grown in complete RPMI-1640 medium (Gibco, Life Technologies, Foster City, CA, USA) supplemented with 10% FBS, MCF10A in a 1:1 mixture of DMEM (Euroclone) and Ham’s F12 (Gibco) supplemented with 10% FBS, 2 μg/mL bovine insulin, 2 ng/mL EGF (insulin and EGF were from Cambrex, East Rutherford, NJ, USA), HaCaT in DMEM (Euro-clone) supplemented with 10% FBS. 16HBE and CFBE cells were grown in MEM medium (Eagle’s Minimal Essential Medium) supplemented with 10% FBS, 2 mM Glutamine (medium and supplements were from Euroclone, Pero, Milan, Italy), and using plastics coated with fibronectin and collagen as previously reported [50]. In all cases a Penicillin–Streptomycin mixture (Euroclone), penicillin G 100 U/mL, streptomycin 0.1 mg/mL, was added to growth media; Fetal Bovine Serum was from Euroclone.

The cells were cultured at 37 °C under 5% CO2 and subculturing was performed according to the supplier’s specification; typically, on arrival the cells were expanded for no more than 10 passages and frozen in aliquots. When required, frozen stocks were thawed using standard procedure and used at passages 4–8 and never after the 10th passage. In no case sign of senescence were observed in growing cells.

When indicated, the cells were treated with 50 nM dexamethasone (Sigma-Aldrich, St. Louis, MO, USA) in culture medium for 6 h.

Nasal brushing from non-CF subjects were obtained by a sterile brush of 2.5 mm thickness.

Leukocytes were isolated from peripheral blood samples using LymphoPrep (Axis-Shield, Dundee, UK) according to the manufacturer’s instructions. In particular, lymphocytes and monocytes were isolated from the interphase of LymphoPrep, while granulocytes were recovered from the residual pellet. To separate lymphocytes from monocytes, an immunomagnetic method was used (Classical monocyte isolation kit, Miltenyi Biotech, Bergisch Gladbach, Germany).

Aliquots of cells and nasal brushing were stored either at −20 °C for DNA extraction or collected in 30 μL of RNA Later (Qiagen, Manchester, UK) and then stored at −80 °C for RNA extraction.

Nasal brushing and leukocytes used in this study were collected during institutional diagnostic procedures. The investigation described here could be carried out on residual specimens following diagnostic analysis with all data kept anonymous. Informed consent to the use was obtained.

### 4.2. RNA Extraction and Expression Analysis

Total RNA was extracted using RNeasy Mini Kit (Qiagen). The RNA samples were treated with DNase I to remove possible contaminating DNA before reverse transcription. For this, 1 μg RNA was treated with 0.4 units DNase I (New England Biolabs, Ipswich, MA, USA) at 37 °C for 10 min; the reaction was stopped by adding 50 mM EDTA at 75 °C for 10 min.

Both a qualitative endpoint PCR and a quantitative real time PCR was performed for the expression analyses of the *SCNN1A*, *SCNN1B*, *SCNN1G*, and *CFTR* genes, as described below.

For the qualitative endpoint PCR, the reverse transcription of 1 μg of RNA was performed using the iScript cDNA Synthesis kit (Bio-Rad, Hercules, CA, USA), according to the manufacturer’s instructions, in a PTC 100 (Biorad, Hercules, CA, USA) PCR system. As also previously described [51,52], the cDNA (2 μL out of 20 μL of the retrotranscription mix) was amplified, using 0.5 U of YieldAce Hotstart DNA polymerase (Stratagene, La Jolla, CA, USA), 175 μM of each dNTP, 1X YieldAce polymerase buffer and 6 pmol of each primer, in a final volume of 15 μL, according to the following PCR protocol: 2 min at 92 °C; from 30 to 40 cycles of 45 s at 94 °C, 1.5 min at 60 °C, 2.5 min at 72 °C; 7 min at 72 °C. The glyceraldehyde 3-phosphate dehydrogenase (*GAPDH)* was used as housekeeping gene. See Appendix A for primers and details. Control PCR reactions included: NRTC (non-retrotranscribed control) in which the cDNA was substituted with total RNA, to identify possible amplification due to genomic DNA; NTC (no template control) containing all components except sample, to identify PCR contamination. Amplicons were analyzed by agarose gel electrophoresis, accompanied by a DNA ladder (GeneRuler 50 bp DNA Ladder, Thermo Scientific, Waltham, MA, USA) made up of 13 fragments with the following sizes (in base pairs): 1000, 900, 800, 700, 600, 500, 400, 300, 250, 200, 150, 100, 50. It contains two high concentration reference bands (500 and 250 bp) for easy orientation.

Quantitative real time PCR was performed using cDNA samples, retro-transcribed with the Reverse Trascription System kit (Promega, Fitchburg, WI, USA) and the PowerUp SYBR Green Master Mix (Applied Biosystems, Forster City, CA, USA) as previously reported [53]. Real time PCR was performed in the ABI7500 Real Time PCR system (Applied Biosystems, Forster City, CA, USA). For each target, the amplification conditions were optimized by running standard curves to ensure that all the targets were amplified with the same efficiency (higher than 95%); additionally, the most efficient concentration of primers was determined by running reactions with different primers concentrations, 300 nM, 600 nM and 900 nM each primer. From these preliminary settings we established the following PCR conditions: 2 μL of cDNA (out of 20 μL of the retrotranscription mix), 1X PowerUp SYBR Green PCR Master Mix (Applied Biosystems, Forster City, CA, USA), 600 nM forward and reverse primers, final volume 25 μL. PCR reactions were set up in triplicate in 96 multi-well plates; NRTC and NTC controls were also included. The cycling conditions were 2 min at 50 °C, 10 min at 95 °C; 40 cycles of 15 s at 95 °C, 1 min at 60 °C, with a final melting curve to control the specificity of amplification products. The *β-actin* was used as housekeeping gene. See Appendix A for primers and details. The relative quantification was calculated by the 2^(−ΔΔC^_T_^)^ method, using the *SCNN1B* expression in CFBE as reference condition. Data are the mean of three independent experiments, each in triplicate.

The specificity and size of the amplicons obtained from qualitative and quantitative expression assays, were also verified by sequencing. For this, samples were run on agarose gels and amplicons were cut out and purified before applying a cycle sequencing protocol on an ABI PRISM 3130*xl* genetic analyzer (Applied Biosystems, Thermo Fisher Scientific, Waltham, MA, USA), using primers reported in Appendix A.

### 4.3. DNA Extraction and Methylation Analysis

Genomic DNA was extracted using QIAamp DNA Mini Kit (Qiagen). Quantification and purity were evaluated spectrophotometrically. DNA methylation was studied by a HpaII/PCR protocol as previously described [54,55]. Briefly, 1 μg of genomic DNA was treated with 3.5 U of the HpaII enzyme (New England Biolabs) in 1X NEBuffer at 37 °C overnight and with additional 2 U of the enzyme at 37 °C for further 6 h. Inactivation was achieved by treatment at 65 °C for 40 min. HpaII cuts CCGG sequence sites only if they are not methylated. After the enzymatic treatment, regions including the CCGG sites of interest were amplified by multiplex PCR starting from 5 ng of genomic digested DNA. The 5′-flanking regions of *SCNN1A* (2926 bp, 7 CCGG sites, 3 target regions, and 1 control region in 1 multiplex Hpa II/PCR), *SCNN1B* (3842 bp, 14 CCGG sites, 4 target regions, and 1 control region in 2 multiplex Hpa II/PCR) and *SCNN1G* (2537 bp, 21 CCGG sites, 2 target regions and 1 control region in 2 multiplex Hpa II/PCR) genes were studied. See Appendix A for primers and details. The HpaII treated genomic DNA was amplified, using 0.5 U of YieldAce DNA polymerase and 1X YieldAce polymerase buffer (Stratagene), 175 μM of each dNTP, 6 pmol of each primer, in a final volume of 15 μL, in a PTC 100 (Biorad) PCR system according to the following touchdown protocol: 2 min at 92 °C; from 30 to 40 cycles of 45 s at 94 °C, 1.5 min starting from 66 °C with a reduction of the Ta of 0.1 °C per cycle, 2.5 min at 72 °C; 7 min at 72 °C.

The results were visualized by agarose gel electrophoresis, accompanied by a DNA ladder (GeneRuler 50 bp DNA Ladder, Thermo Scientific, Waltham, MA, USA), made up of 13 fragments with the following sizes (in base pairs): 1000, 900, 800, 700, 600, 500, 400, 300, 250, 200, 150, 100, 50. It contains two high concentration reference bands (500 and 250 bp) for easy orientation. The presence of the amplicon reveals a region with methylated site(s) (uncut by Hpa II), whereas its absence reveals a region with non-methylated site(s) (cut by Hpa II). The semiquantitative evaluation of the DNA methylation of the studied regions was performed by densitometric assay of amplicons, referred to the amplicon from a control uncut region (contained in each multiplex reaction).

For specificity evaluation and exact sizing, each amplicon (obtained in the Hpa II assay) was sequenced (after recovering from gel) by cycle sequencing on an ABI PRISM 3130*xl* genetic analyzer (Applied Biosystems, Thermo Fisher Scientific, Waltham, MA, USA), using primers reported in Appendix A.

## Figures and Tables

**Figure 1 ijms-22-03754-f001:**
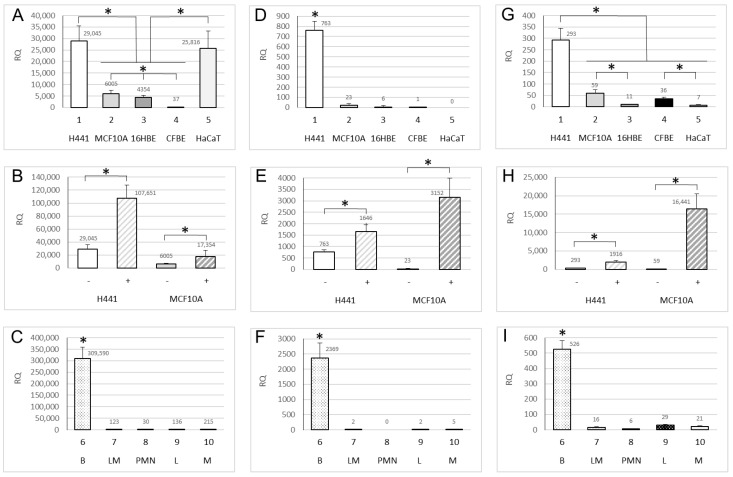
**Expression analysis of ENaC genes by real time PCR.** Results represent the expression of ***SCNN1A*** (**A**–**C**), ***SCNN1B*** (**D**–**F**) and ***SCNN1G*** (**G**–**I**) genes in H441, MCF10A, 16HBE, CFBE, HaCaT cell lines (respectively from 1 to 5, panels **A**,**D**,**G**), H441 and MCF10A with (+) and without (−) dexamethasone treatment (panels **B**,**E**,**H**) and nasal brushing (B), lymphocytes/monocytes (LM), granulocytes (PMN), lymphocytes (L), monocytes (M) (respectively from 6 to 10, panels (**C**,**F**,**I**)). A relative quantification (RQ) is reported on *y*-axis, as fold changes in respect to *SCNN1B* expression in CFBE (panel (**D**), column 4) used as reference condition (the numbers above the bars are the exact RQ values). For panels (**A**,**C**,**D**,**F**,**G**,**I**), ANOVA *p* < 0.01; for panels (**C**,**D**,**F**,**I**) the single * indicates the only statistically significant difference in the panel following Bonferroni’s multiple comparison test (* *p* < 0.01); for panels (**A**) and (**G**) the statistically significant differences between specific conditions following Bonferroni’s multiple comparison test are as indicated (* *p* < 0.01). For panels (**B**,**E**,**H**), Student’s t-test of all dexamethasone treated cells (+) as compared to respective untreated cells (−) * *p* < 0.01.

**Figure 2 ijms-22-03754-f002:**
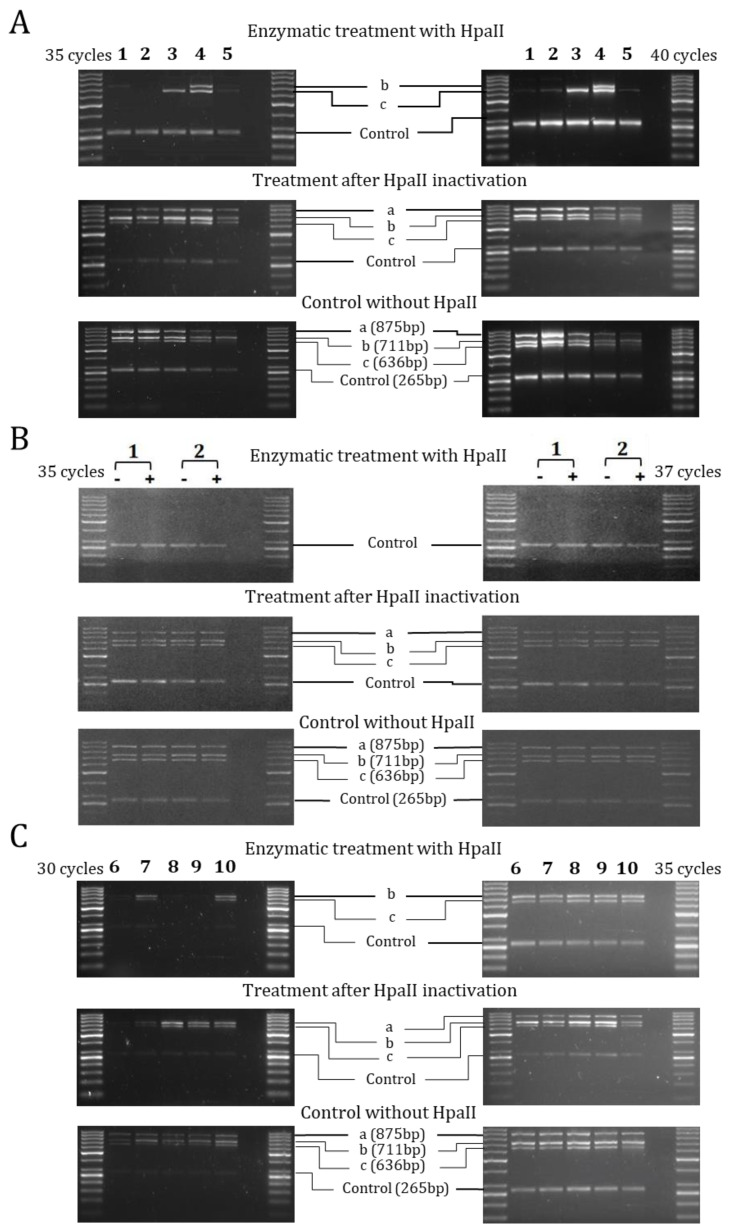
**Methylation analysis of 5′-flanking region of *SCNN1A* gene by Hpa II/PCR.** Results represent products of PCR amplification after HpaII enzymatic treatment of indicated cell lines (1–5, panel (**A**)), H441 and MCF10A with (+) and without (−) dexamethasone treatment (1, 2; panel (**B**)) and ex vivo samples (6–10, panel (**C**)). Each sample was tested at different cycles of PCR amplification protocol, as indicated for panels on the left and on the right of the figure, to better highlight the differences. The analyzed regions are indicated as a, b, c and Control (with the size of amplicons indicated in base pairs (bp) in lower panels). In every panel: 1 = H441, 2 = MCF10A, 3 = 16HBE, 4 = CFBE, 5 = HaCaT, 6 = nasal brushing, 7 = lymphocytes + monocytes, 8 = granulocytes, 9 = lymphocytes, 10 = monocytes. The first and last lane of each panel contain the DNA ladder described in Materials and Methods.

**Figure 3 ijms-22-03754-f003:**
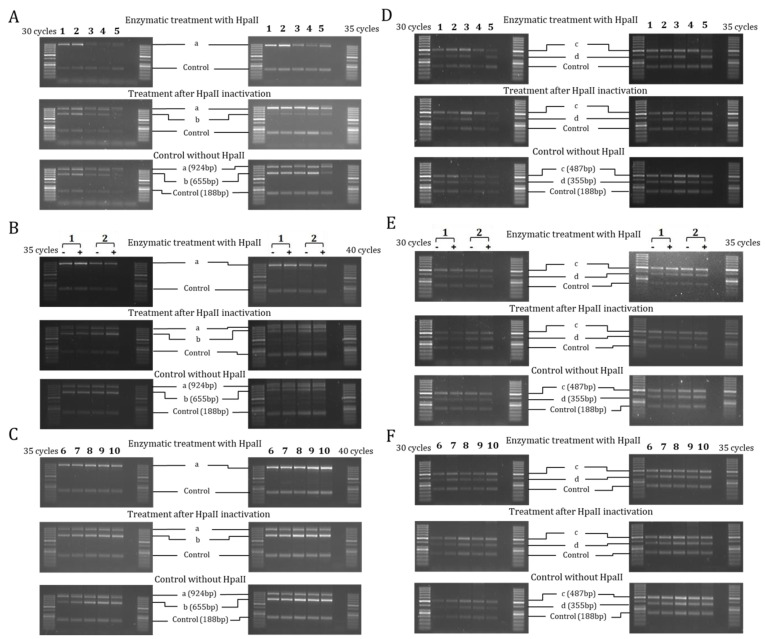
**Methylation analysis of 5′-flanking region of *SCNN1B* gene by Hpa II/PCR.** Results represent products of PCR amplification after HpaII enzymatic treatment of indicated cell lines (1–5, panels (**A**,**D**)), H441 and MCF10A with (+) and without (−) dexamethasone treatment (1, 2; panels (**B**,**E**)) and ex vivo samples (6–10, panels (**C**,**F**)). Each sample was tested at different cycles of PCR amplification protocol, as indicated for panels on the left and on the right of the figure, to better highlight the differences. The analyzed regions are indicated as a, b, c, d and Control (with the size of amplicons indicated in base pairs (bp) in lower panels). In every panel: 1 = H441, 2 = MCF10A, 3 = 16HBE, 4 = CFBE, 5 = HaCaT, 6 = nasal brushing, 7 = lymphocytes + monocytes, 8 = granulocytes, 9 = lymphocytes, 10 = monocytes. The first and last lane of each panel contain the DNA ladder described in Materials and Methods.

**Figure 4 ijms-22-03754-f004:**
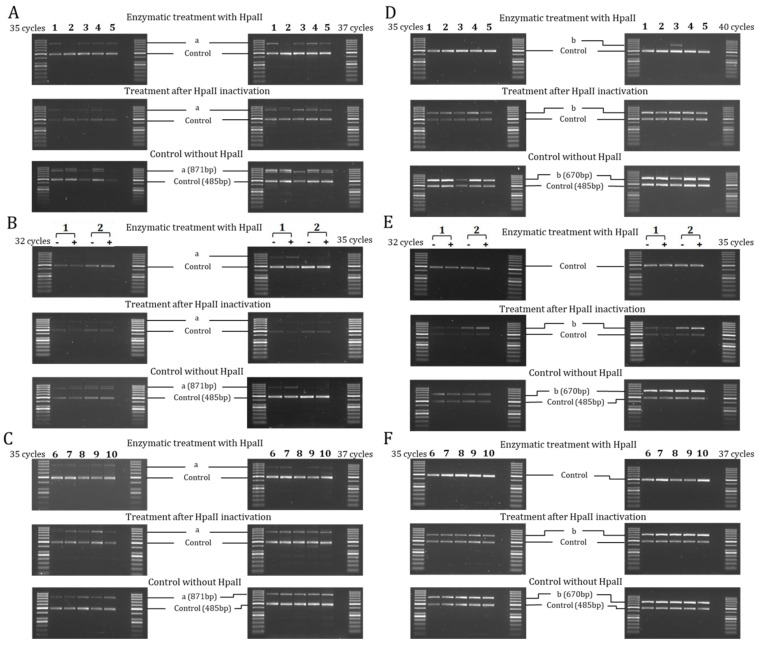
**Methylation analysis of 5′-flanking region of *SCNN1G* gene by HpaII/PCR.** Results represent products of PCR amplification after HpaII enzymatic treatment of indicated cell lines (1–5, panels (**A**,**D**)), H441 and MCF10A with (+) and without (−) dexamethasone treatment (1, 2; panels (**B**,**E**)) and ex vivo samples (6–10, panels (**C**,**F**)). Each sample was tested at different cycles of PCR amplification protocol, as indicated for panels on the left and on the right of the figure, to better highlight the differences. The analyzed regions are indicated as a, b, and Control (with the size of amplicons indicated in base pairs (bp) in lower panels). In every panel: 1 = H441, 2 = MCF10A, 3 = 16HBE, 4 = CFBE, 5 = HaCaT, 6 = nasal brushing, 7 = lymphocytes + monocytes, 8 = granulocytes, 9 = lymphocytes, 10 = monocytes. The first and last lane of each panel contain the DNA ladder described in Materials and Methods.

**Figure 5 ijms-22-03754-f005:**
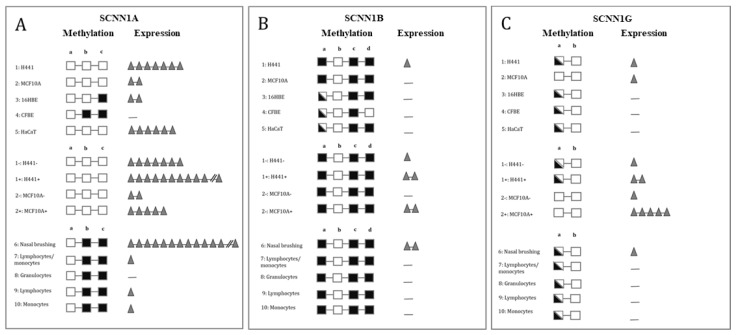
**Schematic representation of results about methylation and expression profiles of ENaC genes.** The profiles of methylation and expression are indicated for cell lines (1–5), H441 and MCF10A (1, 2) with (+) and without (−) dexamethasone treatment, and for ex vivo samples (6–10), as annotated in the legends. Squares represent DNA methylation levels (empty square: unmethylated; half-filled square: partially methylated; filled square: methylated). Triangles represent expression levels (the number of triangles is proportional to the level of expression; no triangle: expression undetectable; one or two triangles: low expression; five to seven triangles: high expression; more than eleven triangles: very high expression). (**A**) Results of the *SCNN1A* gene. a, b, c = methylation analysis regions. (**B**) Results of the *SCNN1B* gene. a, b, c, d = methylation analysis regions. (**C**) Results of the *SCNN1G* gene. a, b = methylation analysis regions.

## Data Availability

Data is contained within the article or Appendix A.

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
