# Peer review of "DNA Methylation Patterns Correlate with the Expression of SCNN1A, SCNN1B, and SCNN1G (Epithelial Sodium Channel, ENaC) Genes"

_ijms, 2021, doi:10.3390/ijms22073754_

Round 1

Reviewer 1 Report

Pierandrei et al. manuscript show evidence that the amount of methylation in the promoter regions of the ENaC channel subunits ?, β and ?, does not regularly control protein expression. Although the results are negative, represent valuable information for other specialists that, in my opinion, are worth publishing and show the complexity of ENaC regulation.   

Reviewer 2 Report

Below are the major concerns and comments I have:

1. Virtually all of the figures are low quality. They are almost the quality of a screen shot from Excel or from the PCR software. There are many details of the figures that cannot be understood if I magnify to analyze the data. Also, it is unclear what everything in the figures stand for. For example, what are the numbers above the bars? What are the ladders or the size of the ladders? 

2. The idea that methylation affects ENaC expression and that this is a possible epigenetic explanation for CFTR regulating ENaC needs to be better described. There are two issues I have with the current approach:

2a. The authors mix methylating ENaC protein with methylating ENaC DNA. The Edinger et al. reference from 2006 for example, examines ENaC protein methylation. This is a long thought but not conclusively demonstrated process by which aldosterone activates ENaC by methylating the protein. This is mixed in with references for methylating ENaC genes as means of turning them down. This needs to be better differentiated and separated and discussed.

2b. Methylating alone of the ENaC genes and examining levels of messenger RNA and cDNA after digesting non-methylated products is fine but what does it mean? Ultimately, this needs to be related to a real change of ENaC protein levels otherwise it is without much significance.

3. The grammar throughout the entire manuscript needs to be improved. It is not scientific quality. This also hold true for referring to some of the processes. For example it is not Bonferroni’s post-test…it is “Bonferroni’s post-hoc test”

4.The abstract is too long and not very informative. There are no details and no numbers summarized in the abstract.  Some things are referred as to high or low or more and less. What do those mean scientifically? Some things like CpG islands are mentioned the last line of the abstract. That should be summarized at the first time methylation is brought up in the abstract.

5. There appears to be a casual almost non-scientific approach to many things. For example “circulating cells”. What are those? Do you mean blood cells? White blood cells? Whatever they are they need to be specifically defined and mentioned and referred to in a correct way. This is just a major example, there are other examples like this. The same rigorous approach needs to be undertaken for all the cells. In terms of isolation, plating, passage number, etc…

6. Missing from all of this are the real time PCR data. All we see is DNA gels which are obviously very limited in their sensitivity. They are also very limited in determining size especially when none of the gels have calibrations and none of the ladders have size listed. It is also not clear what the dynamic range of these gels are? Or even why do we see these and not see the PCR progression curves. Real time PCR has many problems. Without details of the assay, the raw progression curves, internal positive and negative controls it is impossible to evaluate those data.

7. Where are the data for CFTR expression? If CFTR affected ENaC then at least there should be a comparison of changes of CFTR to ENaC. Also, not all CFTR mutants are the same. There needs to be better differentiation between loss of function and loss of trafficking CFTR mutants. Those should have different effects on ENaC- presumably, unless what you are saying of chloride is important. In that case, this is easily tested by changing intracellular chloride.

8. The authors state that not all methylation sites are equivalent. That is fine, but what does that mean? Do some have a larger effect on protein expression? Without those data or at least some link in the literature to that, this would just remain unsubstantiated.
